# Molecular Analysis of SARS-CoV-2 Genetic Lineages in Jordan: Tracking the Introduction and Spread of COVID-19 UK Variant of Concern at a Country Level

**DOI:** 10.3390/pathogens10030302

**Published:** 2021-03-05

**Authors:** Malik Sallam, Azmi Mahafzah

**Affiliations:** 1Department of Pathology, Microbiology and Forensic Medicine, School of Medicine, The University of Jordan, Amman 11942, Jordan; mahafzaa@ju.edu.jo; 2Department of Clinical Laboratories and Forensic Medicine, Jordan University Hospital, Amman 11942, Jordan

**Keywords:** COVID-19, epidemiology, middle-income country, variant, mutation, UK variant, variant of concern

## Abstract

The rapid evolution of severe acute respiratory syndrome coronavirus 2 (SARS-CoV-2) is manifested by the emergence of an ever-growing pool of genetic lineages. The aim of this study was to analyze the genetic variability of SARS-CoV-2 in Jordan, with a special focus on the UK variant of concern. A total of 579 SARS-CoV-2 sequences collected in Jordan were subjected to maximum likelihood and Bayesian phylogenetic analysis. Genetic lineage assignment was undertaken using the Pango system. Amino acid substitutions were investigated using the Protein Variation Effect Analyzer (PROVEAN) tool. A total of 19 different SARS-CoV-2 genetic lineages were detected, with the most frequent being the first Jordan lineage (B.1.1.312), first detected in August 2020 (*n* = 424, 73.2%). This was followed by the second Jordan lineage (B.1.36.10), first detected in September 2020 (*n* = 62, 10.7%), and the UK variant of concern (B.1.1.7; *n* = 36, 6.2%). In the spike gene region, the molecular signature for B.1.1.312 was the non-synonymous mutation A24432T resulting in a deleterious amino acid substitution (Q957L), while the molecular signature for B.1.36.10 was the synonymous mutation C22444T. Bayesian analysis revealed that the UK variant of concern (B.1.1.7) was introduced into Jordan in late November 2020 (mean estimate); four weeks earlier than its official reporting in the country. In Jordan, an exponential increase in COVID-19 cases due to B.1.1.7 lineage coincided with the new year 2021. The highest proportion of phylogenetic clustering was detected for the B.1.1.7 lineage. The amino acid substitution D614G in the spike glycoprotein was exclusively present in the country from July 2020 onwards. Two Jordanian lineages dominated infections in the country, with continuous introduction/emergence of new lineages. In Jordan, the rapid spread of the UK variant of concern should be monitored closely. The spread of SARS-CoV-2 mutants appeared to be related to the founder effect; nevertheless, the biological impact of certain mutations should be further investigated.

## 1. Introduction

The evolutionary analysis of severe acute respiratory syndrome coronavirus 2 (SARS-CoV-2) is appealing for several reasons.

First, this novel virus harbours a ribonucleic acid (RNA) genome, with replication using RNA-dependent RNA polymerase. This replicase enzyme has a minimal proofreading activity; the hallmark of rapidly-evolving viruses (e.g., influenza virus and hepatitis C virus) [1,2].

In addition, the pandemic nature of coronavirus disease 2019 (COVID-19), with more than 100 million detected cases so far, translates into a huge pool of susceptible hosts with varying selective pressure on the viral genome [3,4]. This resulted in rapid divergence of the novel virus from its common ancestor that crossed the species barrier, accompanied by a noticeable genetic diversity in less than a year [5,6,7].

Moreover, the evolutionary analysis of viruses is helpful for epidemiologic purposes [6,8]. Characterization of SARS-CoV-2 genetic variants (monophyletic clades designated as genetic lineages herein) is valuable to track the dynamics of its introduction and dissemination within a certain region [6,9]. Additionally, a consensus on SARS-CoV-2 nomenclature and classification is invaluable for correlating the genetic diversity of the virus with potential biological differences (e.g., antigenicity, transmissibility, virulence) [5,10].

Besides the aforementioned points, phylogenetic studies on SARS-CoV-2 are facilitated by the burgeoning availability of genetic sequences of the virus, as a result of advancement in sequencing technology [5]. As of 3 February 2021; the total number of full-genome SARS-CoV-2 sequences exceeded 400,000 in the global science initiative and primary source for genomic data of influenza viruses (GISAID) [11].

To characterize the genetic lineages of SARS-CoV-2, a comprehensive and dynamic nomenclature/classification scheme was proposed by Rambaut et al. and was referred to as “Pango nomenclature system (from the first-person Latin verb meaning ‘I set’, ‘I fix’ or ‘I record’)” [5,12].

In the Pango system, the following notable lineages have been described so far: (1) lineage A, which dates back to December 2019, and contains the root of COVID-19 pandemic; (2) lineage B, which has Chinese sequences in its base, with branching indicating global exports; (3) lineage B.1, which dates back to January 2020 corresponding to the Italian outbreak; (4) lineage B.1.1.7 (a.k.a. the UK variant of concern) that was detected in the UK in September 2020 with N501Y and P681H as the most notable amino acid substitutions; (5) lineage B.1.351 which dates back to December 2020 (the South African variant) with N501Y, K417N, and E484K as the most notable amino acid substitutions; and (6) lineage P.1 (the Brazilian variant) that was first detected in December 2020 with N501Y, and E484K as the most notable amino acid substitutions [5,13,14,15,16,17].

Amino acid substitutions have been described in SARS-CoV-2 isolates across the world, with possible biological value pending further evidence [18,19,20,21]. Substitutions in the surface (spike) glycoprotein is of particular concern considering its importance in receptor binding and being a target for neutralizing antibodies [22]. The most notable example of such substitutions is the replacement of aspartic acid by glycine at position 614 of the spike glycoprotein of the virus (D614G). The predominance of such substitution was observed in various regions including the Middle East and North Africa [23,24]. Another amino acid substitutions with potential significance include N501Y and E484K in the receptor binding domain of the spike glycoprotein [18,25].

In Jordan, the COVID-19 epidemic went through several phases. In March 2020, mitigation measures took place including travel restrictions, wide lockdowns and curfew, enforcement of masks and social distancing, and prohibition of large gatherings [26,27]. This resulted in clusters of cases during the first five months. However, the inevitable SARS-CoV-2 dissemination started in August/September 2020 when the first wave of community transmission took place [27,28]. Following the peak in active cases and mortality in November 2020, a decline in the number of newly diagnosed cases and deaths was reported in 20 December 2020 and January 2021. The first reporting of B.1.1.7 UK variant of concern dated back to 24 December 2020, and the start of COVID-19 vaccination in the country took place in mid-January 2021 [27]. Several challenges face the country amid the current COVID-19 epidemic including the widespread prevalence of misinformation and conspiracy beliefs, in addition to vaccine hesitancy [26,29,30]. Additionally, the scarcity of molecular and epidemiologic studies on the virus in Jordan can hamper the control efforts in the country.

Thus, the aims of the current study were: (1) to describe the genetic diversity of SARS-CoV-2 in Jordan by analysis of the virus lineages; (2) to analyze the proportion of phylogenetic clustering indicative of local virus spread among the major lineages in the country; and (3) to estimate the timing of the introduction of the UK variant of concern (B.1.1.7) into the country.

## 2. Results

### 2.1. Characteristics of Severe Acute Respiratory Syndrome Coronavirus 2 (SARS-CoV-2) Jordanian Dataset

The total number of GISAID SARS-CoV-2 sequences collected in Jordan and utilized in this study were 579 full-genome sequences. Stratified per month, the number of SARS-CoV-2 sequences collected in the country together with total number of COVID-19 cases and deaths in Jordan are shown in (Table 1). The majority of SARS-CoV-2 sequences were collected in October 2020 (*n* = 344, 59.4%) and September (*n* = 103, 17.8%), which coincided with surge in number of newly diagnosed COVID-19 cases in the country.

### 2.2. Description of the Genetic Lineages of SARS-CoV-2 in Jordan Using the “Pango” System

Using the Pango system, a total of 19 different genetic lineages were found in Jordan. The most common was the first Jordan lineage designated B.1.1.312 (*n* = 424, 73.2%), followed by the second Jordan lineage designated B.1.36.10 (*n* = 62, 10.7%), and the UK variant B.1.1.7 (*n* = 36, 6.2%).

Over a period of 10 months, the predominant lineage shifted from other B lineages during (March–April 2020), to B.1 during (June–July 2020), while the first Jordan lineage B.1.1.312 dominated infections from August 2020 until November 2020, with the concomitant presence of the second Jordan lineage B.1.36.10 over the same period. The last two months (December 2020–January 2021) were dominated by the UK variant B.1.1.7 (Figure 1).

The molecular signature found consistently in the *Spike* gene region of the first Jordan lineage B.1.1.312 was the replacement of adenine by thymine at position 24,432 (A24432T) of the reference genome NC_045512 (thymine instead of uracil since the results were those of DNA sequencing). This mutation was non-synonymous resulting in the replacement of glutamine (Q) by leucine (L) at position 957 of the spike glycoprotein (Q957L).

The molecular signature in the *Spike* gene region for the second Jordan lineage B.1.36.10 was C22444T (a synonymous mutation).

Using the Tamura–Nei model, the evolutionary divergence for both B.1.1.312 and B.1.36.10 from the reference SARS-CoV-2 sequence NC_045512, was 0.00064 substitutions/site, while the divergence from the reference sequence was the highest for B.1.1.7 (0.00188 substitutions/site). Assessing within-lineage genetic diversity using the same model revealed the highest diversity within B.1.36.10 sequences (0.00023 substitutions/site, and within B.1.1.312 sequences (0.00022 substitutions/site), while the genetic diversity was the lowest among the UK lineage B.1.1.7 (0.00008 substitutions/site).

### 2.3. The Proportion of Phylogenetic Clustering among the Three Most Common Lineages in Jordan

To determine sequence clustering among the three most common genetic lineages of SARS-CoV-2 in Jordan, we conducted maximum likelihood (ML) phylogeny construction. Using the *Spike* gene region, the proportion of phylogenetic clustering was the highest among the B.1.1.7 lineage sequences (35/35, 100.0%) followed by B.1.36.10 sequences (19/62, 30.6%), and B.1.1.312 sequences (126/414, 30.4%).

A higher proportion of phylogenetic clustering for the Jordan lineages was detected using the open reading frame 1ab (*ORF1ab*) region, with the highest proportion of clustering also seen among the lineage B.1.1.7 sequences (32/35, 91.4%), followed by B.1.36.10 (44/62, 71.0%) and B.1.1.312 (176/401, 43.9%, Figure 2). Please refer to the Materials and Methods section for the explanation of difference in B.1.1.312 number of sequences for the two sub-genomic regions (Appendix A).

### 2.4. Amino Acid Substitutions in the Surface Glycoprotein of the Three Major Genetic Lineages in Jordan

For the three major genetic lineages in Jordan (B.1.1.312; B.1.36.10 and B.1.1.7), an assessment of amino acid substitutions in the spike glycoprotein compared to that in the reference sequence (YP_009724390) was undertaken.

The amino acid substitution D614G was detected in the vast majority of sequences (*n* = 566, 97.8%), and the wild type (D614) was last identified in June 2020.

The amino acid substitutions N501Y and P681H besides the deletion Δ69/70 were consistently found among the lineage B.1.1.7 sequences, while N501I was detected in a single sequence from the first Jordan lineage B.1.1.312.

The following amino acid substitutions were totally absent from the sequences that were analyzed in this study: K417N and E484K.

Using the Protein Variation Effect Analyzer (PROVEAN) tool, two amino acid substitutions were predicted to be deleterious for the spike glycoprotein: T716I detected among B.1.1.7 sequences and Q957L found in the first Jordan lineage B.1.1.312 (Table 2).

### 2.5. The UK Variant of Concern was Introduced into Jordan in Late November 2020

Bayesian analysis of the UK variant of concern (B.1.1.7) lineage, with 35 SARS-COV-2 *S* sequences collected in Jordan between 24 December 2020 and 6 January 2021 revealed that the time to the most recent common ancestor (tMRCA) of this lineage in Jordan was 21 November 2020 (95% highest posterior density interval: 17 November 2020–24 December 2020). Coalescent analysis using a Bayesian skyline plot showed a rapid exponential increase in the number of effective infections between 1 January 2021 and 5 January 2021 (Figure 3).

## 3. Discussion

In this study, we utilized molecular clock and coalescent analyses to describe the timeline of introduction of the genetic lineage B.1.1.7—commonly known as the UK variant of concern—and its spread in Jordan. Additionally, we employed the Pango classification system, which facilitates the classification and nomenclature of SARS-CoV-2 genetic lineages, containing molecular signatures that can be helpful to track its introduction/emergence and spread [5]. This approach can be used to evaluate public health measures including control and mitigation practices [31]. The negative consequences of the current COVID-19 pandemic necessitates such in-depth epidemiologic studies, which can be helpful to plan effective preventive strategies [32,33].

The major result of this study revealed that the genetic lineage B.1.1.7 was introduced into Jordan about four weeks earlier than the official reporting of its introduction into the country [27]. Bayesian skyline coalescent analysis showed that the exponential increase in infections as a result of the B.1.1.7 lineage coincided with the new year 2021, following a lag phase of several weeks. It is known that the human behavior can drive a surge in infections if a super spreader event takes place in a large gathering [34,35]. However, this hypothesis needs further evaluation using contact tracing data together with dense sampling to reconstruct the evolutionary history of this lineage in the country.

Despite the need for further evidence regarding the biological significance of B.1.1.7 lineage, several studies reported on the rapid dissemination of this lineage in UK among several other countries [6,16,36,37]. This proposed change in virus behavior can be related to enhanced binding between the spike glycoprotein of this lineage and its receptor; and this enhancement has been proposed to be the result of N501Y amino acid substitution [18,38].

Additionally, we used the Pango classification system to describe the molecular epidemiology of COVID-19 in Jordan [5]. Since the first introduction of the novel coronavirus into humans, the expanding genetic diversity of the virus demanded a scheme to classify and name monophyletic clades, which would facilitate the study of epidemiologic features of the virus including its spread. This would also provide a consensus to study the possible biological significance of such lineages [39,40]. In this study, we adopted the approach conceived by Rambaut et al., that can help in analyzing patterns of introduction and spread of this novel virus in a certain region [5,12].

Community transmission of SARS-CoV-2 in Jordan became apparent in August 2020, and was dominated by three genetic lineages starting with the first and second Jordan lineages (B.1.1.312 and B.1.36.10), and it was recently driven by the UK variant of concern (B.1.1.7.). The emergence/introduction of the two Jordan lineages can be mostly related to a founder effect, since no discernible advantageous or neutral mutations were detected among the two lineages [41,42,43]. The molecular signature of the second Jordan lineage (B.1.36.10) was found in earlier sequences collected in Turkey [44]. This might point to a possibility of introduction of this lineage into Jordan in early September 2020, considering that travelers coming from Turkey (classified as a green country at that time) were not required to be quarantined [45].

One result that should be investigated further is the higher proportion of phylogenetic clustering for the B.1.1.7 lineage compared to the two Jordan lineages. This indicates a higher proportion of domestic transmission, which can be linked to enhanced transmissibility of the virus. However, such a result is pending further evidence to support the current observations linking such a genetic lineage with a higher transmission [37].

In line with several previous studies, genetic analysis of SARS-CoV-2 in Jordan showed the shift into B lineage, harboring the spike D614G amino acid substitution, with all sequences collected in Jordan harboring this substitution from July 2020 onwards [23,24]. This amino acid substitution was present in the country as early as March 2020, which hints to the effects of virus genetic changes on its epidemic behavior, despite the need for further evidence to support such a correlation [21,46,47,48].

### Study Strengths and Limitations

The current study used the state-of-the-art phylogenetic inference methods to characterize the molecular epidemiology of SARS-CoV-2 in Jordan. Additionally, this study can be considered among the first studies in the Middle East and North Africa region utilizing the Pango classification system to characterize the genetic diversity of SARS-CoV-2 to the best of our knowledge.

Limitations of this study included potential sampling bias in time, which was manifested by variation in sequencing proportion in relation to new cases diagnosed each month; with 1.3% sequencing rate out of the newly diagnosed cases before October 2020 and 0.1% thereafter.

Another caveat of this study can be the enhanced surveillance of passengers (and their contacts) coming from UK or other countries where the UK variant of concern was reported. This may have caused the dominance of B.1.1.7 lineage among sequences collected in December 2020–January 2021.

## 4. Materials and Methods

### 4.1. Compilation of SARS-CoV-2 Jordanian Dataset and Epidemiologic Data

All SARS-CoV-2 genetic sequences that were collected in Jordan were retrieved from GISAID, as of 30 January 2021 [11]. The Jordanian sequences were aligned together with the reference SARS-CoV-2 sequence Wuhan-Hu-1 (accession number: NC_045512). Multiple sequence alignment was undertaken through a multiple alignment program for amino acid or nucleotide sequences (MAFFT v.7) [49].

Data on daily COVID-19 diagnosed cases and deaths in Jordan were retrieved from Coronavirus Source Data, and covered the period from 3 March 2020 to 29 January 2021 [50].

### 4.2. SARS-CoV-2 Lineage Assignment

To describe the genetic lineages of the sequences in the SARS-CoV-2 Jordanian dataset, we utilized Phylogenetic Assignment of Named Global Outbreak Lineages (Pangolin) [51]. The Pangolin tool follows the ‘Pango’ nomenclature system for classifying SARS-CoV-2 genomic sequences [5,12].

The measurement of within-lineage genetic distances was done using MEGA6, which was also used to detect the following amino acid substitutions/deletions in the spike glycoprotein sequence: D614G, E484K, N501Y, P681H, 69–70del, and K417N [52].

Genetic divergence from the reference sequence of SARS-CoV-2 and within-lineage genetic diversity were assessed using the Tamura–Nei model as implemented in MEGA6 [52,53].

### 4.3. Assessment Spike Protein of the Major Lineages in Jordan

For the three major SARS-CoV-2 lineages circulating in Jordan (B.1.1.312; B.1.36.10; and B.1.1.7), we used the Protein Variation Effect Analyzer (PROVEAN) tool in order to assess the possible functional changes in the spike glycoprotein compared to that in the reference sequence (YP_009724390) [54].

### 4.4. Maximum Likelihood Phylogenetic Analysis

To conduct the ML phylogenetic analysis, we used two sub-genomic parts of the dataset: (1) *ORF1ab* (NC_045512 positions: 266–21,555); with the following Jordanian sequences removed for having long (>10) stretches of ambiguous (N) bases: EPI_ISL_429992; EPI_ISL_429995; EPI_ISL_430008; EPI_ISL_430013; EPI_ISL_450189; EPI_ISL_636390; EPI_ISL_730391; EPI_ISL_730473; EPI_ISL_730545; EPI_ISL_755118; EPI_ISL_755120; EPI_ISL_755121; EPI_ISL_755122; EPI_ISL_755123; EPI_ISL_755124; EPI_ISL_755125; EPI_ISL_755126; EPI_ISL_755127; EPI_ISL_755128; EPI_ISL_755129; EPI_ISL_755131; EPI_ISL_755237; EPI_ISL_755238; EPI_ISL_755239; EPI_ISL_755240; EPI_ISL_755243; EPI_ISL_755247; EPI_ISL_755267; EPI_ISL_878495; which yielded a dataset with 550 Jordanian sequences; (2) Spike *S* (NC_045512 positions: 21,563–25,384); with the following Jordanian sequences removed for having long (> 10) stretches of ambiguous (N) bases: EPI_ISL_430013; EPI_ISL_450189; EPI_ISL_755118; EPI_ISL_755120; EPI_ISL_755121; EPI_ISL_755123; EPI_ISL_755125; EPI_ISL_755126; EPI_ISL_755128; EPI_ISL_755131; EPI_ISL_878495; EPI_ISL_430008; EPI_ISL_730543; EPI_ISL_730545; which yielded a dataset with 565 Jordanian sequences.

Phylogeny construction for the two sub-genomic Jordanian datasets using the ML approach was done using PhyML v3 [55]. The Smart Model Selection (SMS) was used for selection of the most appropriate nucleotide substitution model, depending on the Akaike Information Criterion (AIC) [56]. Models that were used for construction of ML trees were: GTR + G for *ORF1ab*; and HKY85 + I for *S* region.

### 4.5. Timing of B.1.1.7 Lineage (UK Variant) Introduction into Jordan

To estimate the time to the most recent common ancestor (tMRCA) for the B.1.1.7 lineage in Jordan, we used the Bayesian Markov chain Monte Carlo (MCMC) method in BEAST v1.8.4 [57]. The following criteria were used for Bayesian evolutionary analysis by sampling trees (BEAST) analysis: HKY nucleotide substitution model with discrete gamma-distributed rate heterogeneity, uncorrelated relaxed clock model with a uniform rate prior (initial value of 0.0065) and a Bayesian skyline tree density model [23]. A single run with 200 million chain length was performed, with samples of trees and parameters collected every 20,000 steps after discarding a burn-in of 20%. Convergence was checked for using Tracer v1.6.0. with all parameters having effective sample sizes (ESSs) of >200. Construction of the Bayesian skyline plot was done in Tracer; and assembly of the maximum clade credibility (MCC) tree was done using TreeAnnotator available in BEAST package [57]. Visualization of the trees in this study was undertaken in FigTree [58].

## 5. Conclusions

In the current study, molecular characterization of SARS-CoV-2 in Jordan was undertaken for the first time to the best of our knowledge. A recent report by Edyth Parker et al investigated the emergence of lineage B.1.1.7 in Jordan and revealed the current dominance of this lineage in Jordan [59]. Two Jordan lineages dominated infections in the country, with a recent introduction of the lineage B.1.1.7. This UK variant of concern was present in the country several weeks before its official reporting, with an exponential propagation over the first few days of the new year 2021.

The introduction of new lineages in the country appeared to be related to founder effect; nevertheless, the biological significance of certain mutations should be further evaluated. An important note should be clarified, which is related to the distinction that should be made between the epidemiologic and contact tracing value of determination of virus lineages as opposed to the identification and characterization of novel strain, subtypes or types of viruses that have distinct biological features. Thus, continuous surveillance of genetic variability of SARS-CoV-2 is recommended to track the emergence of new genetic variants, with subsequent studies of its potential biological significance.

The media hype about the UK variant of concern seems justified considering its rapid spread and the number of amino acid changes detected in the spike glycoprotein of this lineage, which can have important effects on antigenicity and transmissibility. In turn, this can have implications for the current vaccine formulations and resurgence of new waves of infection.

## Figures and Tables

**Figure 1 pathogens-10-00302-f001:**
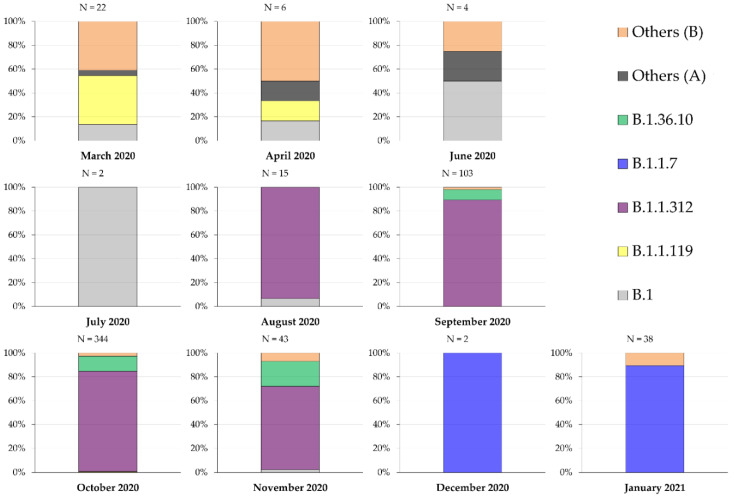
Distribution of SARS-CoV-2 genetic lineages over the period (March 2020–January 2021). Others (**A**) include the following lineages: A and A.5; Others (**B**) include the following lineages: B.1.1.1, B.1.1.114, B.1.1.227, B.1.1.51, B.1.2, B.1.311, B.1.319, B.1.36, B.1.457, B.28, B.35 and B.40.

**Figure 2 pathogens-10-00302-f002:**
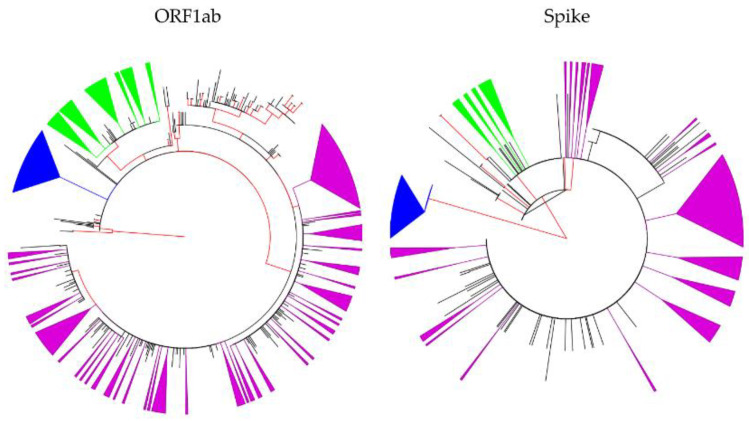
Maximum likelihood (ML) phylogenetic trees constructed using SARS-CoV-2 sequences collected in Jordan. The ML tree to the left was constructed using *ORF1ab* region, while the ML tree to the right was constructed using the *Spike* region. Internal branches with approximate likelihood Shimodaira–Hasegawa (aLRT-SH) values of ≥0.90 are shown in red. The clustered first Jordan lineage (B.1.1.312) sequences are shown as collapsed purple triangles; the clustered second Jordan lineage (B.1.36.10) are shown as collapsed green triangles; and the clustered UK variant lineage (B.1.1.7) are shown as collapsed blue triangles; ORF: open reading frame.

**Figure 3 pathogens-10-00302-f003:**
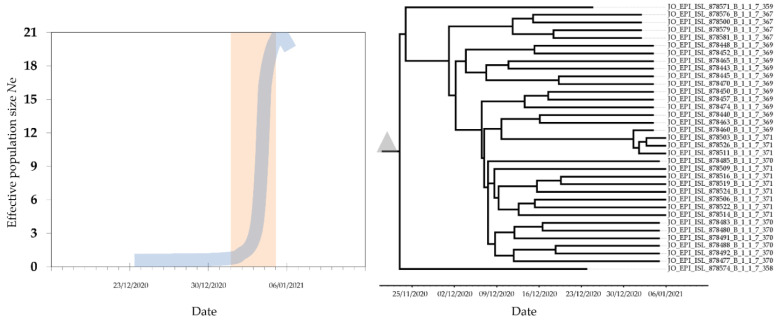
Maximum clade credibility (MCC) tree of the lineage B.1.1.7 (UK variant of concern) in Jordan, with the mean estimate for the tMRCA shown as the grey triangle (**right**). The median effective population size (*N*e) shown in blue displayed a lag phase in December 2020 followed by an exponential increase in infections starting on 1 January 2021 highlighted in orange rectangle (**left**).

**Table 1 pathogens-10-00302-t001:** The total number of coronavirus disease 2019 (COVID-19) new cases, deaths and severe acute respiratory syndrome coronavirus 2 (SARS-CoV-2) sequences in Jordan (March 2020–January 2021).

Month	Newly Diagnosed COVID-19 ^1^ Cases	COVID-19 Related Deaths	Number of SARS-CoV-2 ^2^ Sequences	Percentage of Sequences Compared to New Cases
March 2020	274	5	22	8.0292%
April 2020	179	3	6	3.3520%
May 2020	286	1	0	0
June 2020	393	0	4	1.0178%
July 2020	61	2	2	3.2787%
August 2020	841	4	15	1.7836%
September 2020	9791	46	103	1.0520%
October 2020	60,782	768	344	0.5660%
November 2020	146,823	1922	43	0.0293%
December 2020	75,064	1083	2	0.0027%
January 2021	30,539	447	38	0.1244%

^1^ COVID-19: Coronavirus disease 2019; ^2^ SARS-CoV-2: Severe acute respiratory syndrome coronavirus 2.

**Table 2 pathogens-10-00302-t002:** Prediction of amino acid substitution impact in the spike glycoprotein of SARS-CoV-2 stratified by the three major genetic lineages detected in Jordan.

SARS-CoV-2 Lineage	Amino Acid Substitution	PROVEAN ^1^ Score	Prediction (Cutoff = −2.5)
UK variant of concern (B.1.1.7)	H69_V70del	0.808	Neutral
	V143_Y144del	1.318	Neutral
	N501Y	−0.090	Neutral
	A570D	−0.682	Neutral
	D614G	0.598	Neutral
	P681H	0.060	Neutral
	T716I	−3.293	Deleterious
	S982A	−1.505	Neutral
	D1118H	−1.142	Neutral
First Jordan lineage (B.1.1.312)	D614G	0.598	Neutral
	Q957L	−2.929	Deleterious
Second Jordan Lineage (B.1.36.10)	D614G	0.598	Neutral

^1^ Variants with a score equal to or below −2.5 are considered “deleterious,” and variants with a score above −2.5 are considered “neutral” in the Protein Variation Effect Analyzer (PROVEAN) tool.

## Data Availability

The data authors can be contacted directly via GISAID website: https://www.gisaid.org/. The da-tasets analysed during the current study (ML analyses files, xml files without sequences, Tracer log files) are available from the corresponding author (M.S.) on a reasonable request and considering the terms of use by GISAID.

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
