# Peer review of "Molecular Analysis of SARS-CoV-2 Genetic Lineages in Jordan: Tracking the Introduction and Spread of COVID-19 UK Variant of Concern at a Country Level"

_pathogens, 2021, doi:10.3390/pathogens10030302_

Round 1

Reviewer 1 Report

The manuscript presents some interesting data on the SARS-CoV-2 genetic lineages in Jordan as one of the first studies in this region. The subject is of great interest in the medical world.

Some improvements could be made to the manuscript:

  • The title could be reconsidered as not only the COVID-19 UK variant was analyzed in the study.
  • I would structure the manuscript on the classic basis: BAckground, Material and methods, results, and conclusions.
  • The Introductions seem too long compared to the Discussion. I would move some data from the introduction to the discussion.
  • The presentation of the variation of the genetic lineages over the study period could be better arranged by month and not as pies, but columns bar chart.
  • In the Discussion, the authors could introduce a paragraph about the study's strengths, not only the limitations.
  • I would reduce the conclusions' length; some sentences from there can be moved to the Discussions.
  • There is no mention of the ethical approval of the study.

There is no need for an English language check as it sounds good; also, the study is clearly presented, and data from the Introduction and Discussions are based on a large and recent list of references.

Author Response

Reviewer #1 comments

The manuscript presents some interesting data on the SARS-CoV-2 genetic lineages in Jordan as one of the first studies in this region. The subject is of great interest in the medical world.

Some improvements could be made to the manuscript:

  1. The title could be reconsidered as not only the COVID-19 UK variant was analyzed in the study.

Response: We would like to thank the reviewer for this important comment. The main focus of this study was on timing the introduction and spread of the UK variant of concern; thus, we highlighted this in the title. But to highlight the molecular epidemiology analysis in this study, we changed the title accordingly into “Molecular Analysis of SARS-CoV-2 Genetic Lineages in Jordan: Tracking the Introduction and Spread of COVID-19 UK Variant of Concern at a Country Level”.

  1. I would structure the manuscript on the classic basis: BAckground, Material and methods, results, and conclusions.

Response: We would like to thank the reviewer for this comment.

In the preparation of this manuscript, we followed the guidelines presented in the Pathogens Journal website and the also we used the Journal template which ordered the sections as follows:  Manuscript Preparation

General Considerations

Research manuscripts should comprise:

Front matter: Title, Author list, Affiliations, Abstract, Keywords

Research manuscript sections: Introduction, Results, Discussion, Materials and Methods, Conclusions (optional).

Back matter: Supplementary Materials, Acknowledgments, Author Contributions, Conflicts of Interest, References.

Thus, we would like to keep the structure of the manuscript in the current form that follows the journal guidelines presented here, which stated that “Authors must use the Microsoft Word template or LaTeX template to prepare their manuscript.”: https://www.mdpi.com/journal/pathogens/instructions#preparation

  1. The Introductions seem too long compared to the Discussion. I would move some data from the introduction to the discussion.

Response: We would like to thank the reviewer for giving us the opportunity to enhance the readability and flow of text in our manuscript. Based on the reviewer’s comment, we moved one paragraph from the introduction section into the discussion as follows:

In the Introduction section (Page 2, Lines 62-65) were moved into the Discussion section (Page 7, Lines 207-211).

  1. The presentation of the variation of the genetic lineages over the study period could be better arranged by month and not as pies, but columns bar chart.

Response: We would like to thank the reviewer for this important comment, that helped us to make this Figure clearer. Based on the reviewer’s suggestion we replaced the pie charts with column bar charts to show the changes in genetic lineages of SARS-CoV-2 over the study period in Jordan.

Please refer to the revised highlighted manuscript, Page 4.

  1. In the Discussion, the authors could introduce a paragraph about the study's strengths, not only the limitations.

Response: Based on the reviewer’s comment, we added the following paragraph to the Discussion section: “Study strengths and limitations

The current study used the state-of-the-art phylogenetic inference methods to characterize the molecular epidemiology of SARS-CoV-2 in Jordan. Additionally, this study can be considered among the first studies in the Middle East and North Africa region utilizing the Pango classification system to characterize the genetic diversity of SARS-CoV-2 to the best of our knowledge.”

  1. I would reduce the conclusions' length; some sentences from there can be moved to the Discussions.

Response: We would like to thank the reviewer for this suggestion; however, we believe that keeping the conclusions in the current format would help to deliver the main conclusions of this study.  

  1. There is no mention of the ethical approval of the study.

Response: We would like to thank the reviewer for this important comment, and as mentioned in the backmatter of the manuscript, the IRB approval is not applicable since we used an already published molecular sequences of the virus that are publicly available on the open database GISAID.

  1. There is no need for an English language check as it sounds good; also, the study is clearly presented, and data from the Introduction and Discussions are based on a large and recent list of references.

Response: We would like to thank the reviewer for the insightful and thorough review of this manuscript.

Reviewer 2 Report

The authors demonstrated the SARS-COV-2in Jordan by selecting 579 full-genome sequences for analysis, their results demonstrated that there are three main frequent lineages presented in Jordan, and later information about these three lineages is better analyzed.

I agree with the contents which the authors illustrated, only have some suggestions for the author to improve the manuscript.

1.The abstract is a little bit too long, maybe try to restructure a bit.

  1. In Fig.1 about the distribution of SARS-COV-2 lineage B.1.1.312, the authors were using the kind of orange color in the graph, but in Fig2, with the same lineage, the authors were using purple color, in my opinion, it's nicer to keep the same color for indications from the beginning until the end. 

3.In Fig2, the lineage information should be better added in the figure, and as the picture is too big to put in the manuscript, it's nicer to include the sequence information and also together with the original phylogenetic tree in the supplementary file.

4.line200, fig2 should be fig3

Author Response

Reviewer #2 comments

The authors demonstrated the SARS-COV-2in Jordan by selecting 579 full-genome sequences for analysis, their results demonstrated that there are three main frequent lineages presented in Jordan, and later information about these three lineages is better analyzed.

I agree with the contents which the authors illustrated, only have some suggestions for the author to improve the manuscript.

  1. The abstract is a little bit too long, maybe try to restructure a bit.

Response: We would like to thank the reviewer for the comment. Accordingly, we made the following changes to the abstract:

We deleted the following: “Genetic characterization of SARS-CoV-2 is used to track lineage emergence/introduction and spread in a certain region.” And “Pangolin COVID-19 Lineage Assigner”.

  1. In Fig.1 about the distribution of SARS-COV-2 lineage B.1.1.312, the authors were using the kind of orange color in the graph, but in Fig2, with the same lineage, the authors were using purple color, in my opinion, it's nicer to keep the same color for indications from the beginning until the end.

Response: We would like to thank the reviewer for the important comment. Accordingly, and based on the reviewer’s suggestion, we changed the color scheme of Figure 1 to be consistent with Figure 2 color for the lineages and we used the purple color to indicate B.1.1.312 lineage.

Please refer to the edited Figure 1, in Page 4 of the revised highlighted manuscript.

  1. In Fig2, the lineage information should be better added in the figure, and as the picture is too big to put in the manuscript, it's nicer to include the sequence information and also together with the original phylogenetic tree in the supplementary file.

Response: We would like to thank the reviewer for this important comment. Accordingly, we added a supplementary file (Supplementary S1), with detailed maximum likelihood phylogenetic trees for the Jordanian SARS-CoV-2 sequences including sequence information.

  1. line200, fig2 should be fig3.

Response: We would like to thank the reviewer for this comment, and we corrected this typographical error according to the reviewer’s comment.

Reviewer 3 Report

In the manuscript “Analysis of SARS-CoV-2 Genetic Lineages in Jordan: Tracking the Introduction and Spread of COVID-19 UK Variant of Concern at a Country Level”, the author aimed to investigate the genetic variability of SARS-CoV-2 in Jordan, with a special 14 focus on the UK variant of concern.

 The topic of the manuscript is certainly interesting and captures one of the issues of the moment on the spread of infection during the Sars-Cov-2 pandemic, in light of new scientific evidence on the topic.  

However, there are some points that need further clarification:

  1. The structure of the manuscript appears confusing. In fact, the authors followed an unconventional order in writing the paper, placing the results after the introduction, then the discussion, and deferring the materials and methods as the penultimate paragraph. This structure makes for confusing reading. Please deeply restructure the text.

  1. Please briefly discuss the impact of COVID on daily life around the world, on the sport activity, immune system and cardiovascular disease by citing recent review on the topic. (ref. Exercise, Immune System, Nutrition, Respiratory and Cardiovascular Diseases during COVID-19: A Complex Combination - Int J Environ Res Public Health. 2021 Jan 21;18(3):904)

  1. Please, you should explain each of your abbreviations the first time it appears in the main text (i.e. RNA) and provide a description of the abbreviations for figures in the caption (ie. ORF, etc).

Author Response

Reviewer #3 comments

In the manuscript “Analysis of SARS-CoV-2 Genetic Lineages in Jordan: Tracking the Introduction and Spread of COVID-19 UK Variant of Concern at a Country Level”, the author aimed to investigate the genetic variability of SARS-CoV-2 in Jordan, with a special 14 focus on the UK variant of concern.

 The topic of the manuscript is certainly interesting and captures one of the issues of the moment on the spread of infection during the Sars-Cov-2 pandemic, in light of new scientific evidence on the topic. 

However, there are some points that need further clarification:

  1. The structure of the manuscript appears confusing. In fact, the authors followed an unconventional order in writing the paper, placing the results after the introduction, then the discussion, and deferring the materials and methods as the penultimate paragraph. This structure makes for confusing reading. Please deeply restructure the text.

Response: We would like to thank the reviewer for the important comment. As mentioned in our response to reviewer #1 second point, we followed Pathogens Journal submission guidelines and we used the template available to authors at: https://www.mdpi.com/journal/pathogens/instructions

Thus, we followed the Journal instructions stating that the Research manuscript sections: Introduction, Results, Discussion, Materials and Methods, Conclusions (optional).

  1. Please briefly discuss the impact of COVID on daily life around the world, on the sport activity, immune system and cardiovascular disease by citing recent review on the topic. (ref. Exercise, Immune System, Nutrition, Respiratory and Cardiovascular Diseases during COVID-19: A Complex Combination - Int J Environ Res Public Health. 2021 Jan 21;18(3):904)

Response: We would like to thank the reviewer for this comment, and based on this comment we added a short paragraph in the discussion to highlight the importance of studying the epidemiology of COVID-19 considering the negative impact of this pandemic on the different aspects of life as mentioned in the suggested reference, which was added to the reference list as follows:

Discussion section (Page 7, Lines 211-213): “The negative consequences of the current COVID-19 pandemic necessitates such in-depth epidemiologic studies, which can be helpful to plan effective preventive strategies [32,33].”

References:

  1. Scudiero, O.; Lombardo, B.; Brancaccio, M.; Mennitti, C.; Cesaro, A.; Fimiani, F.; Gentile, L.; Moscarella, E.; Amodio, F.; Ranieri, A., et al. Exercise, Immune System, Nutrition, Respiratory and Cardiovascular Diseases during COVID-19: A Complex Combination. Int J Environ Res Public Health 2021, 18, doi:10.3390/ijerph18030904.
  2. Park, M.; Cook, A.R.; Lim, J.T.; Sun, Y.; Dickens, B.L. A Systematic Review of COVID-19 Epidemiology Based on Current Evidence. J Clin Med 2020, 9, doi:10.3390/jcm9040967.
  3. Please, you should explain each of your abbreviations the first time it appears in the main text (i.e. RNA) and provide a description of the abbreviations for figures in the caption (ie. ORF, etc).

Response: We would like to thank the reviewer for this comment, and accordingly, we made the following changes:

Page 1, line 40: ribonucleic acid (RNA),

Page 5, line 169: ORF: open reading frame.

We also added a Table of abbreviations in the end of the manuscript (Page 10, Lines 367-389).

Round 2

Reviewer 1 Report

The authors improved the manuscript by responding to some of the recommendations.

The abstract can be organized in the classic structure: background, aim, methods, results, conclusions.

Still, if it is possible, I would combine the 10 images of Figure 1 in one image with all the bars in the same row.

Author Response

The authors improved the manuscript by responding to some of the recommendations.

The abstract can be organized in the classic structure: background, aim, methods, results, conclusions.

Still, if it is possible, I would combine the 10 images of Figure 1 in one image with all the bars in the same row.

Response: We would to thank the reviewer for this comment, but again we followed the guidelines presented in the Pathogens Journal website https://www.mdpi.com/journal/pathogens/instructions#preparation

For Figure 1, we think it would be clearer to keep it in the current format.

Reviewer 2 Report

I have no more comments on the revised version except for the color in figure1 and figure2, for me, it looks like they are not exactly the same color, could the authors make them exactly the same?

Author Response

Reviewer #2 comments

I have no more comments on the revised version except for the color in figure1 and figure2, for me, it looks like they are not exactly the same color, could the authors make them exactly the same?

Response: We would like to thank the reviewer for the comment, but we prefer to keep the color format in the current form.

Reviewer 3 Report

Please specify "ORF" abbrevation in the main text. 

The authors responded fully to the reviewers' concerns. 
Therefore, there are no further comments. 

Author Response

Reviewer #3 comments

Please specify "ORF" abbrevation in the main text.

The authors responded fully to the reviewers' concerns.

Therefore, there are no further comments.

Response: We would like to thank the reviewer for the comment, and accordingly we edited it in the manuscript (Page 5, line 157)